# Extensional fault geometry and evolution within rifted margin hyper-extended continental crust leading to mantle exhumation and allochthon formation

Júlia Gómez-Romeu[1,*] & Nick Kusznir[1]

[1]*Department of Earth, Ocean and Ecological Sciences, University of Liverpool, Liverpool, UK* [*]*Currently:* M&U sasu, *Sassenage, France*

## Abstract

Seismic reflection interpretation at magma-poor rifted margins shows that crustal thinning within the hyper-extended domain occurs by in-sequence oceanward extensional faulting which terminates in a sub-horizontal reflector in the top-most mantle immediately beneath tilted crustal fault blocks. This sub-horizontal reflector is interpreted to be a detachment surface which develops sequentially with oceanward in-sequence crustal faulting. We investigate the geometry and evolution of active and inactive extensional faulting due to flexural isostatic rotation during magma-poor margin hyper-extension using a recursive adaptation of the rolling hinge model of Buck (1988) and compare modelling results with published seismic interpretation. In the case of progressive in-sequence faulting, we show that sub-horizontal reflectors imaged on seismic reflection data can be generated by the flexural isostatic rotation of faults with initially high-angle geometry. Our modelling supports the hypothesis of Lymer et al. (2019) that the S reflector on the Galician margin is a sub-horizontal detachment generated by the in-sequence incremental addition of the isostatically rotated soles of block bounding extensional faults. Flexural isostatic rotation produces shallowing of emergent fault angles, fault locking and the development of new high-angle short-cut fault segments within the hanging-wall. This results in the transfer and isostatic rotation of triangular pieces of hangingwall onto exhumed fault footwall, forming extensional allochthons which our modelling predicts are typically limited to a few km in lateral extent and thickness. The initial geometry of basement extensional faults is a long-standing question. Our modelling results show that a sequence of extensional listric or planar faults with otherwise identical tectonic parameters produce very similar sea-bed bathymetric relief but distinct Moho and allochthon shapes. Our preferred interpretation of our modelling results and seismic data is that faults are initially planar in geometry but are isostatically rotated and coalesce at depth to

form the seismically observed sub-horizontal detachment in the top-most mantle. In-sequence extensional faulting of hyper-extended continental crust results in a smooth bathymetric transition from thinned continental crust to exhumed mantle; in contrast out-of- sequence faulting results in a transition to exhumed mantle with bathymetric relief.

## 1. Introduction

The formation of a rifted continental margin during continental breakup requires continental crust and lithosphere to be stretched and thinned. In the case of a magma-poor rifted margins, 5 progressive stages of margin formation resulting in 5 distinct margin domains have been identified: proximal, necking, hyper-extended, exhumed mantle and oceanic crust (Mohn et al. 2012, Tugend et al. 2014). The hyper-extended domain of a magma-poor rifted margin forms when the crust is thinned to approximately 10 km thickness or less and the crust becomes fully brittle allowing faults to penetrate through the entire crust into the mantle (Pérez-Gussinyé et al., 2001; Manatschal, 2004). The hyper-extended domain has a crustal architecture characterised by tilted crustal fault blocks separated by oceanward dipping basement extensional faults and often underlain by a strong sub-horizontal seismic reflector. This is illustrated on figure 1(a) which shows a seismic reflection dip section (Lymer et al. 2019) within the hyper-extended domain of the distal Galicia Bank margin west of Iberia. The sub-horizontal reflector, known as the S reflector, has been interpreted to be a sub-horizontal detachment within the top-most mantle (Krawczyk et al., 1996; Reston et al., 1996) into which basement extensional faults sole.

The geometry and evolution of extensional faults and their relationship to the S reflector within the hyper-extended domain have been a long-standing question. Interpretation of 2D seismic reflection data (Ranero and Pérez-Gussinyé, 2010) has revealed that basement extensional faulting within the hyper-extended domain develops oceanward in-sequence with new faults developing in the oceanward direction at the same time as abandonment of earlier faults. Recent high-quality 3D seismic reflection seismic on the SW of Galicia Bank west of Iberia (Lymer at al. 2019) confirms this oceanward in-sequence fault development and additionally provides observations that determine the relationship between the in-sequence basement extensional faulting and the underlying S sub-horizontal reflector. Basement extensional faults are observed to sole out into the sub-horizontal detachment within the top-most mantle imaged as the S seismic reflector. In 3D the S reflector shows corrugations that indicate the direction of slip and correlate with corrugations within the extensional block-bounding faults. Further analysis by Lymer at al. (2019) reveals that the S reflector is a composite surface made by the progressive ocean-ward in-sequence development of a sub-horizontal detachment into which

the higher angle basement faults sole. Their analysis also reveals that as extension migrates oceanward in-sequence and that several faults may be active simultaneously. A similar relationship has been observed between basement extensional faulting and sub-horizontal S type seismic reflectors in other rift basins using 3D seismic reflection data. Figure 1(b) shows corrugations on the sub-horizontal reflector interpreted as a detachment surface and its relationship to basement extensional faulting above for the Porcupine Basin west of Ireland (Lymer at al. 2022). Lymer et al. (2019) present a schematic summary (reproduced in Figure 1c) of extensional basement faulting in the hyper-extended domain and its relationship to the sub-horizontal detachment within the top-most mantle, most probably controlled by serpentinization, into which basement faults sole.

Dynamic thermo-rheological finite element models of continental lithosphere stretching and thinning (e.g. Lavier & Manatschal, 2006; Brune et al. 2014; Naliboff et al. 2017) leading to continental breakup and rifted margin formation have been successful in simulating the progression from necking to hyper-extension to mantle exhumation at magma-poor rifted margins. However these dynamic models do not replicate the extensional fault and detachment structures observed on 2D and 3D seismic reflection data. The dynamic model of Peron-Pinvidic & Naliboff (2020), specifically investigating extensional detachment development, predicts extensional fault structures that penetrate to depths much greater than the seismically observed S-type reflector; additionally their predicted fault geometries remain steep failing to match the lower fault angles imaged on seismic reflection data. The kinematic model presented by Ranero & Perez-Gussinye (2010) using extensional fault block rotation much better replicates extensional fault and detachment structures imaged by 2D seismic within the hyper-extended magma-poor margin domain. Their work however preceded the 3D seismic observations by Lymer et al (2019) of the S-type detachment and its corrugations.

Lymer et al. (2019) propose that their observations strongly support the development of the S seismic reflector by a rolling-hinge process (Buck 1988) in which a sub-horizonal detachment is created by the incremental addition of the soles of basement extensional faults. The kinematic rolling-hinge model of Buck (1988) has been successfully used at slow-spreading ocean ridges to replicate and analyse extensional faulting leading to footwall exhumation, detachment faulting and core complex formation (Smith et al. 2008; Schouten et al, 2010). In this paper, we use a recursive adaptation of the rolling hinge model of Buck (1988) to examine how both active and inactive fault geometries are modified by flexural isostatic rotation during sequential faulting to form the sub-horizonal structure imaged on seismic reflection data.

A long-standing question is whether the initial geometry of crustal extension faults is planar or listric; earthquake seismology and geodetic observations favour a planar geometry (Jackson 1987; Stein & Barrientos 1985). Using the flexural isostatic rotation model, we also investigate whether an initial listric or planar fault geometry better fits seismic observations of the sub-horizontal reflector and the geometry of extensional allochthons. In addition, we examine the transition from hyper-extended continental crust to exhumed mantle and how it depends on the sequence of extensional faulting.

## 2. Model formulation

We use a numerical model (RIFTER) to replicate faulting and fault block geometry within the hyper-extended domain, and to investigate fault rotation, fault geometry interaction, the formation of crustal allochthon blocks and the transition between hyper-extended and exhumed mantle domains. RIFTER is a kinematic forward lithosphere deformation model that allows the production of flexural isostatically compensated as well as balanced cross-sections. Within RIFTER, lithosphere is deformed by faulting in the upper crust with underlying distributed pure-shear deformation in the lower crust and mantle. RIFTER can be used to model and predict the structural development in extensional tectonic settings as shown in Figure 2. The model is kinematically controlled with fault geometry, fault displacement and pure-shear distribution given as model inputs as a function of time.

The kinematic formulation of RIFTER represents an advantage over dynamic modelling because the input data given to RIFTER can be constrained by observed geology. Specifically fault position, extension magnitude and sequence order with respect to other faults can be taken directly from the interpretation of seismic reflection images and used to drive the kinematic model. This is in contrast to dynamic models where fault location, extension magnitude and sequence order are predicted by the model and may bare little relationship to an observed structural and stratigraphic cross-section. In a kinematic model, while the lithosphere deformation is specified as an input, the thermal and isostatic consequences may be dynamically determined to predict thermal uplift and subsidence (e.g. Gómez-Romeu et al. 2019). Because model outputs are geological cross-sections which are flexural isostatically compensated as well as structurally balanced, RIFTER provides for the isostatic testing of palinspastic cross-sections and can also be used to explore different kinematic scenarios. A more detailed description of the model formulation (originally called OROGENY) is given by Toth et al., (1996), Ford et al., (1999) and Jácome et al., (2003). These studies show the model formulation applied to compressional tectonics however similar physical principles apply for an extensional tectonics scenario. Gómez-Romeu et al., (2019) show how RIFTER can be used to reproduce both extensional and compressional tectonics using the Western Pyrenees as a case-study.

Within RIFTER, loads resulting from extensional lithosphere deformation are compensated by flexural isostasy. These loads are generated by faulting, crustal thinning, sedimentation, erosion and lithosphere thermal perturbation and re-equilibration (Kusznir et al., 1991). The lithosphere flexural strength must be considered to determine the isostatic rotation of faults during extension and therefore to investigate their geometric evolution. For the purposes of calculating the flexural isostatic response, the lithosphere is represented as an elastic plate of effective elastic thickness (Te) floating on a fluid substratum. The lithosphere effective elastic thickness (Te) is defined as the equivalent thickness of a perfectly elastic plate which has the same flexural strength as the lithosphere. Extension on basement faults produces flexure which, as well as generating footwall uplift and hangingwall subsidence, gives rise to substantial bending stresses (Magnavita et al., 1994) in the cooler upper lithosphere; these large bending stresses are reduced by combined brittle and plastic failure. The flexural strength of the lithosphere, and therefore Te, are reduced by this brittle and plastic failure and this reduction becomes greater with increase in extension (Magnavita et al., 1994). Therefore, in extensional tectonic settings, a low effective elastic thickness (Te) is expected and required to reproduce the consequences of lithosphere deformation due to extensional faulting.

We use a Te value of 0.5 km in our modelling of extensional faulting during the formation of the hyperextended domain and mantle exhumation (Figure 3). This value is consistent with those determined at slow-spreading ocean ridges ranging between 0.5 and 1 km (e.g. Buck, 1988; Smith et al., 2008; Schouten et al., 2010) where a similar lithosphere flexural strength to that of the distal rifted margins is expected. The sensitivity of model predictions to Te is shown in Figure 4; increasing Te increases the bathymetric relief resulting from extensional faulting but otherwise the structural architecture remains similar.

The initial crustal geometry for our modelling of extensional faulting within the hyperextended domain leading to mantle exhumation and allochthon formation is when the continental crust has been thinned down to 10 km (Tugend et al., 2014) corresponding to the point when faults within the seismogenic layer couple into the mantle (Pérez-Gussinyé et al., 2001). Prior to that, during the necking zone stage of margin formation (Mohn et al., 2012), faults are expected to be decoupled from the mantle by ductile deformation within the lower continental crust. The width of the necking zone with crust 10 km thick at the start of hyperextension is set to 100 km although this width value is not critical to this study. The starting bathymetry is set to 2 km corresponding to the isostatic equilibrium of continental crust thinned to 10 km with an highly elevated lithosphere geotherm (Figure 3b). For simplicity we only model faulting during hyper-extension on one distal rifted margin and do not include faulting within its distal conjugate. This simplified initial model template allows us to focus on extensional faulting

during the hyper-extension stage of magma-poor rifted margin formation avoiding the complexity occurring during the earlier rifting and necking phases. Figure 3c shows the resultant model of a hyper-extended distal rifted margin. The detailed numerical model stages to produce this are shown in Figures 3d-e and described below for the formation of the hyperextended domain, the initiation of the exhumed mantle domain and the formation of extensional allochthons.

## 3. Model application to sequential faulting within the hyper-extended margin domain

The interpretation of sub-horizontal seismic reflectors below fault blocks within the hyperextended domain has been intensively debated (e.g. Reston et al., 1996). Interpretations suggested for the S-type reflectors on the Iberian margin (de Charpal et al., 1978; Krawczyk et al., 1996) are many and are reviewed later in the discussion. Despite this wide range of possible interpretations, after the work by Reston et al. (1996) and Krawczyk et al. (1996), it has been generally accepted that the S-type reflectors are detachment faults (Manatschal et al., 2001). Ranero & Pérez-Gussinyé (2010) show that extensional faulting within the hyper-extended domain develops oceanward insequence with initially steeply dipping faults. As in-sequence faulting propagates oceanward, active fault rotation modifies the deeper geometry of previously active faults leading to their deeper segments being passively rotated to a lower angle producing an apparent listric fault geometry or even a sub-horizontal appearance. Lymer et al., (2019) confirmed observationally that extensional faulting develops oceanward in-sequence, and that extensional faulting soles out into the sub-horizontal detachment imaged as the S-type-reflectors.

Figure 3d shows the modelling results of progressive deformation within the hyper-extended domain resulting from a set of in-sequence extensional faults. The initial pre-movement dip of each extensional fault at the surface is 60°. This value is consistent with Andersonian extensional fault mechanics (Anderson 1905) and also the value of 55° – 60° determined for initial surface fault dip by Lymer et al. (2019) from their analysis of 3D seismic reflection data on the SW Galicia Bank margin. Note that our RIFTER modelling results shown in this paper, using high initial faults angles, do not apply to low angle extensionally reactivated thrusts (Morley, 2009; Deng et al. 2022).

In the model results shown in Figure 3d-e the faults detach at 15 km depth corresponding to an assumed brittle-plastic transition within the topmost mantle (results obtained from an initial planar fault geometry are examined later). Flexural isostatic response to faulting leads to an uplift of the footwall block, subsidence of the hanging-wall block and a rotation of the active fault plane reducing its dip

(Figure 3d1). The reduction of fault dip due to flexural isostatic rotation is expected to lead to the locking of that fault and the initiation of new faults with steeper dip. This is shown in Figure 3d2 and subsequent Figures 3d3-6.

Extension on each new fault not only reduces its own fault dip by flexural isostatic rotation but also further reduces the fault dip of earlier active faults within its footwall. The cumulative result of this process is that faults originally steeply dipping when active become sub-horizontal in their lower parts as illustrated in Figures 3d5 for fault number 1. In this case the sub-horizontal inactive fault is almost coincident with the Moho beneath the hyper-extended continental crustal fault-blocks (Figure 3d5). If fault extension is sufficiently large and the hyper-extended continental crust is sufficiently thin, footwall exhumation leads to mantle exhumation (Figure 3d6) as proposed by Manatschal et al. (2001).

Table 1 summarizes the fault parameters and sequential fault displacement required to reproduce the structural architecture of the hyper-extended domain shown in Figure 3d.

## 4. Model application to mantle exhumation and extensional allochthon formation

For even greater extension on the exhumation fault, the exhumed mantle footwall becomes sub-horizontal at the sea-bed due to flexural isostatic rotation as predicted by the rolling-hinge model of Buck (1988). Extensional allochthon blocks sitting above sub-horizontal exhumed footwall are observed at magma-poor margins by seismic reflection imaging and field studies (Epin and Manatschal and references therein, 2018).

We use RIFTER to investigate the formation of extensional allochthon blocks by the rolling hinge model as suggested by Manatschal et al., (2001) and shown in Figure 3e. Allochthon blocks are produced by new steeply dipping extensional faults cutting through the hangingwall block of a master fault (fault 6 in our case in Figure 3e1) and pulling off triangular pieces of continental crust from the hanging-wall (i.e. the rolling hinge model of Buck, 1988). These new faults, created when the emergence angle of the master fault becomes too low (~30° dip), are short-cuts of the master fault and connect with it at depth. Depending on what depth they initiate at and their break-away position, the size of the crustal allochthon block generated will vary (Figure 3e). The intersection depth between the master fault and the new extensional faults is different in each model stage shown in Figure 3e but it ranges between 5 and 10 km depth consistent with deMartin et al., (2007). Another parameter that differs in each model stage is the distance between two consecutive allochthon blocks. This depends on how much the new extensional fault moved before it locked. A small fault offset will not generate

exhumed mantle between two allochthon blocks as shown in Figures 3e3-4 whereas a large fault offset will generate exhumed mantle and a sub-horizontal sea-bed geometry between two allochthon blocks (Figures 3e4-5). Note that each allochthon block overlies sub-horizontal exhumed footwall generated by flexural isostatic rotation.

Table 2 summarizes the initial fault parameters and the chronological fault displacement required to reproduce the structural architecture of the exhumed mantle domain shown in Figure 3e.

The RIFTER model results shown in Figure 3 do not include sediment deposition during hyper-extension, mantle exhumation and allochthon formation. In Figure 5, incremental sediment deposition and its isostatic loading are included in the model; the tectonics remains the same as in Figure 3. The model results of increasing sediment supply are shown in Figures 5b-c and compared with the model result with no sediment deposition shown in Figure 5a. Because of the diachronous tectonics of oceanward in-sequence extensional faulting during the formation of the distal magma-poor margin, sediments of the same age may be syn-tectonic if they are deposited where active faulting is occurring, or they may be post-tectonic if they are passive fill of accommodation space generated by earlier extensional faulting that has ceased at that location. The important distinction between syn- and post-tectonic sedimentation due to diachronous tectonics during rifted margin formation is described in greater detail in Ribes et al. (2019) and Manatschal et al (2022).

Figure 5b shows a relatively small amount of sediment incrementally added to the model and is consistent with a relatively sediment starved scenario corresponding to the SW Galicia margin as imaged by the 3D seismic of Lymer et al (2019). The isostatic response to the small amount of sediment loading shown in Figure 5b is also small and the flexural isostatic fault rotation is therefore not significantly different from the model result with no sediments shown in Figure 5a. The increased isostatic response to increasing sediment supply (Figures 5c&d) results in a slight decrease in fault rotation resulting in slightly steeper faults for the same fault extension. Sediment supply and its isostatic loading are therefore expected to exert a control on when faults lock and new oceanward in-sequence faults develop.

## 5. Sensitivity to listric or planar fault geometry?

Lithosphere deformation is achieved by localised deformation on faults and shear zones within the upper lithosphere with distributed deformation below at depth. A long-standing question is how deformation by faulting connects to deep distributed lithosphere deformation. This question also has implications for fault geometry. Our numerical experiments described above in sections 3 and 4 assume

a listric fault geometry in which faults sole out into a sub-horizontal shear zone at 15 km depth below which deformation becomes distributed. In contrast earthquake seismology and geodetic analysis (Stein and Barrientos, 1985; Jackson, 1987) suggests that large extensional earthquakes involve faults whose geometry is planar.

We explore the differences between using listric and planar fault in modelling the formation of the hyper-extended and exhumed mantle domains. The results are compared in Figure 6. The initial faults geometries for listric and planar faults are shown in Figures 5a and d respectively. Both have an initial surface dip of 60°. The initial listric fault geometry soles out at 15 km while the initial planar fault geometry continues downwards with a dip of 60°. We assume that the deformation transition from faulting to distributed deformation for the planar fault occurs within the mantle below the crust-mantle density interface and so does not affect the isostatic response to faulting.

Listric and planar fault geometry model predictions are shown in Figures 6c and f and use the same fault locations, fault extension and sequence. Comparison shows that listric and planar fault geometries produces very similar sea-bed structural topography, and which cannot be used to distinguish whether fault geometry is listric or planar. In contrast, the listric and planar fault models produce different sub-surface structure. The Moho geometries predicted by the listric and planar fault geometry models are also different, however whether these different predicted Moho geometries can be distinguished using seismic reflection data is uncertain.

In section 4 we used listric fault geometries to model allochthon formation. We now examine allochthon formation using planar faults and compare these predictions with those using listric faults (Figure 7). For both listric and planar fault geometries, Figure 7 shows the formation of allochthons for different separations of the hanging-wall short-cut fault from the primary extensional fault which has exhumed mantle footwall. Separations of 1 km (Figures 7a-b and g-h), 2 km (Figures 7c-d and i-j) and 5 km (Figures 7 e-f and k-l) are used. For the 1 km separation, a small allochthon is produced with similar triangular geometry for both listric (Figure 7b) and planar (Figure 7h) fault geometries. Increasing the separation to 2 km increases the allochthon size; however while the listric fault (Figure 7d) produces a triangular allochthon, the planar fault (Figure 7j) geometry produces a 4-sided body. For a 5 km separation, the allochthon size increases further and both listric (Figure 7f) and planar (Figure 7l) fault geometries produce a 4- sided body. For the larger separations of the short-cut fault from the primary fault, the detached fragment transferred to the exhumed mantle consists of continental basement with some autochthonous mantle beneath it (Figure 7j-l). Whether extensional

allochthons can provide insight into answering the question are extensional faults listric or planar poses an interesting challenge.

## 6. The transition from hyper-extended crust to exhumed mantle and its sensitivity to in-sequence vs out-of-sequence faulting

Stretching and thinning of the continental crust can eventually lead to mantle exhumation as observed by drilling on the distal Iberian margin (Figures 8a-b). Seismic reflection data (Figure 8c) provides insight into how mantle exhumation was achieved by extensional faulting. Based on drill and seismic reflection data, Manatschal et al., (2001, 2004) proposed that an in-sequence ocean-ward propagating set of extensional faulting progressively thins the continental crust in the hyper-extended domain until eventually a large extensional fault exhumes mantle in its footwall. Our modelling of mantle exhumation using a set of in-sequence extensional faults as proposed by Manatschal et al., (2001, 2004) is shown in Figure 3 and 9a and produces a smooth bathymetric transition from continental crust to exhumed mantle.

While the in-sequence fault extension process provides a very good generalised model for the formation of the hyper-extended margin domain, mantle exhumation and their transition, it is unlikely that all faults propagate in-sequence oceanward. Some out-of-sequence faulting is to be expected when the 3D nature and along strike complexity of rifting and breakup is considered and can be seen seismically in Figure 8e. In Figure 9b we show the result of introducing an out-of-sequence fault, with the same dip sense as other faults, into the hyperextension and mantle exhumation model. All other faults have similar locations and extensions to those used to produce Figure 9a. The effect of introducing an out-of-sequence fault to exhume mantle is to produce a transition from thinned continental crust to mantle which is no longer smooth at the seabed but shows bathymetric relief. An out-of-sequence fault might also have an opposite dip-sense as shown in Figure 9c. This fault does not exhume mantle but does generate a horst containing exhumed mantle capped by thinned continental crust as observed in Figure 8e.

## 7. Discussion

To better understand extensional fault geometry and its evolution during hyper-extension at magma-poor rifted margins, several important questions need to be answered: (i) are faults active at low angle, (ii) what is the relationship between the sub-horizontal reflector and block bounding faults, (iii) do faults have a listric or planar geometry and (iv) is faulting always in-sequence.

In section 4 (Figure 3) we show for a listric fault geometry that flexural isostatic rotation progressively reduces the fault dip of inactive faults within the footwall of oceanward in-sequence faulting. From this we can deduce that the present-day sub-horizontal orientation of a fault at depth does not indicate that the fault was active at a sub-horizontal orientation. This conclusion is consistent with the modelling results of Ranero & Pérez-Gussinyé, (2010) and the 3D seismic observations of Lymer et al. (2019).

The nature of the seismically Imaged sub-horizontal reflectors beneath rotated fault blocks in the hyper-extended domain has been extensively debated (e.g. Reston et al. 1996; Lymer et al. 2019 and references therein). Proposed origins of the sub-horizontal reflector have included a lithosphere scale extensional detachment fault (Wernicke et al., 1981), the top of a mafic underplate (Horsefield, 1992), a thin igneous intrusion (Reston, 1996), a serpentinization front (Boillot et al., 1987), and the brittle-plastic transition (de Charpal et al., 1978; Sibuet, 1992). Detailed seismology by Reston et al., (1996) was able to eliminate an igneous origin, leaving a sub-horizontal detachment in the top-most mantle as the most likely interpretation, probably assisted by mantle serpentinization (Pérez Gussinyé et al., (2001).

Seismic reflection interpretation shows that extensional faults thinning the continental crust within the hyper-extended domain sole out into the sub-horizontal reflector (Reston et al. 1996; Manatschal et al., 2001). If extensional faults within the hyper-extended zone penetrate into the mantle, as suggested by Pérez Gussinyé et al., (2001), then the interpretation of seismically observed sub-horizontal reflectors being a sub-horizontal detachment requires it to be within the mantle rather than at the base of the thinned continental crust. Analysis of the recently acquired 3D seismic reflection data in the hyper-extended southern Galicia margin by Lymer et al. (2019) shows that oceanward in-sequence extensional crustal faulting detaches into a sub-horizontal detachment imaged as the sub-horizontal reflector (confirming the interpretations of Manatschal et al.; 2001 and Ranero & Pérez-Gussinyé: 2010). Their 3D analysis of the correlation between corrugations within the S reflector surface and those within block bounding faults demonstrates that the sub-horizontal detachment imaged as the S reflector develops synchronously with the oceanward in-sequence crustal faulting.

Our listric fault model (Figure 3a-c) assumes that faults sole out into a horizontal detachment within the top-most mantle consistent with the seismically observed sub-horizontal S reflector being interpreted as a horizontal detachment into which the block bounding extensional faults above sole into. Our model is also consistent with the interpretation of Lymer et al., (2019) that the sub-horizontal reflector is the relict of an oceanward propagating detachment at the base of the in-sequence crustal

faulting and is not simultaneously active from distal to proximal. Our modelling supports the hypothesis of Lymer et al. (2019) that the S reflector on the Galicia margin is a sub-horizontal detachment generated by the in-sequence incremental addition of the isosatically rotated soles of block bounding extensional faults.

In section 5 (Figure 6) we compare the response of listric and planar fault geometries for oceanward in-sequence hyper-extension. Significant flexural isostatic rotation leading to greatly reduced dip of planar faults at depth is also seen, especially for planar faults in the footwall of later faults with large extension. However, Figure 6 shows a clear difference between planar (Figures 6d-f) and listric (Figures 6a-c) fault geometries at depth; planar fault geometries do not result in a continuous sub-horizontal structure at depth. In contrast because all listric faults sole out at the same brittle-plastic transition depth, all listric faults form a single continuous sub-horizontal structure at depth resembling that observed on seismic reflection data in the hyper-extended domain.

Earthquake seismology, however, favours a planar fault geometry for extension within the seismogenic layer (Stein and Barrientos, 1985; Jackson, 1987). How might extensional deformation on a planar fault in the brittle seismogenic layer terminate at depth? In the case of rifted margin hyper-extension, faults penetrate the crust and permit water to penetrate down into the top-most mantle (e.g. Pérez-Gussinyé et al., 2001) enabling mantle serpentinization to occur. Serpentinized top-most mantle at the base of extensional faults would produce a weak layer enabling the formation of a horizontal detachment. Planar faulting in the seismogenic layer, isostatically rotated to low angles, would then sole out into this horizontal detachment in the top-most serpentinised mantle immediately beneath thinned continental crust. The resulting fault geometry would not be dissimilar to that of the listric fault used in the modelling of sections 3 and 4 but with a more planar geometry in the upper brittle seismogenic layer as observed on the 3D seismic of Lymer et al. (2019).

The rolling hinge model of Buck (1988) provides an explanation for the formation of triangular allochthons of continental crust emplaced on exhumed mantle (Buck 1988; Manatchal et al. 2001; Epin & Manatschal, 2019). In Figures 3 and 7 we show slivers of hanging wall continental crust transferred onto exhumed mantle footwall by short-cut faults. Flexural isostatic rotation produces the observed geometry of triangular allochthons emplaced on sub-horizontal exhumed mantle. While listric and planar fault geometries produce nearly identical small allochthons, their difference becomes pronounced for large allochthons (Figure 7). Listric faults always produce a triangular allochthon fragment of hanging-wall continental crust while planar faults produce a rectangular shape for large allochthons (semantically these large rectangular fragments produced by planar faults should perhaps

be called autochthons). Whether reflection seismology observations of large allochthon shapes can be used to distinguish listric or planar fault geometry during hyper-extension remains to be investigated.

Oceanward in-sequence faulting shown in Figure 3 and as proposed by Manatschal et al. (2001) and Manatschal (2004) provides a good generalised model for the formation of hyper-extended magma-poor margins. However, it should be recognised that out-of-sequence faulting does occur during margin formation and is the inevitable consequence of the 3D nature of continental breakup at the regional scale where upper-plate/lower-plate polarity varies along margin strike. Lymer et al., (2019) also show that, at the more local scale, 3D fault system overlap must occur and would also break a simple oceanward in-sequence fault pattern. The transition from hyper-extended continental crust to exhumed mantle is particularly sensitive to the sequence of faulting; oceanward in-sequence faulting produces a smooth bathymetric transition onto exhumed mantle while out of sequence produces a transition with bathymetric relief as shown in Figure 9.

## 8. Summary

a) Flexural isostatic rotation of extensional faulting (the rolling hinge model) applied to the formation of the hyper-extended domain of magma-poor rifted margins predicts fault geometry evolution consistent with the published interpretations of 3D seismic reflection data.

b) The same modelling shows that seismically observed low-angle extensional faults were not necessarily active at low angle and have been flexurally rotated to their present low angle geometry.

c) Modelling supports the hypothesis of Lymer et al. (2019) that the S reflector on the Galicia margin is a sub-horizontal detachment generated by the in-sequence incremental addition of the isostatically rotated soles of block bounding extensional faults.

d) Extensional faults may initially have a planar geometry in the upper seismogenic layer but this initial planar geometry is modified by flexural isostatic rotation.

e) The predicted geometry of extensional allochthons emplaced on exhumed mantle is sensitive to the initial geometry of block bounding faults. This may provide a means of distinguishing listric and planar faults using seismic reflection data.

f) Sequential in-sequence oceanward extensional faulting is the dominant process during the extensional thinning of the hyper-extended domain at magma-poor rifted margins. Some out-of-sequence faulting does occur and generates a recognisably distinct transition onto exhumed mantle.

## Author contribution

**JGR**: Conceptualization, Formal analysis, Investigation, Methodology, Visualization, Writing – original draft preparation, Writing – review and editing. **NK**: Conceptualization, Formal analysis, Funding acquisition, Investigation, Methodology, Project administration, Software, Supervision, Visualization, Writing – review and editing.

## Competing interests

The authors declare that they have no conflict of interest.

## Acknowledgments

We thank the MM4 (Margin Modelling Phase 4) industry partners (BP, Conoco Phillips, Statoil, Petrobras, Total, Shell, BHP-Billiton, and BG) for financial support. We also thank Tony Dore &Chris Morley for constructive reviews and Alan Roberts and Gianreto Manatschal for discussions. We also thanks Gael Lymer for his assistance with seismic images used in Figure 1.

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

Iberia ocean-continent transition and ophiolites. Tectonics 12, 5.

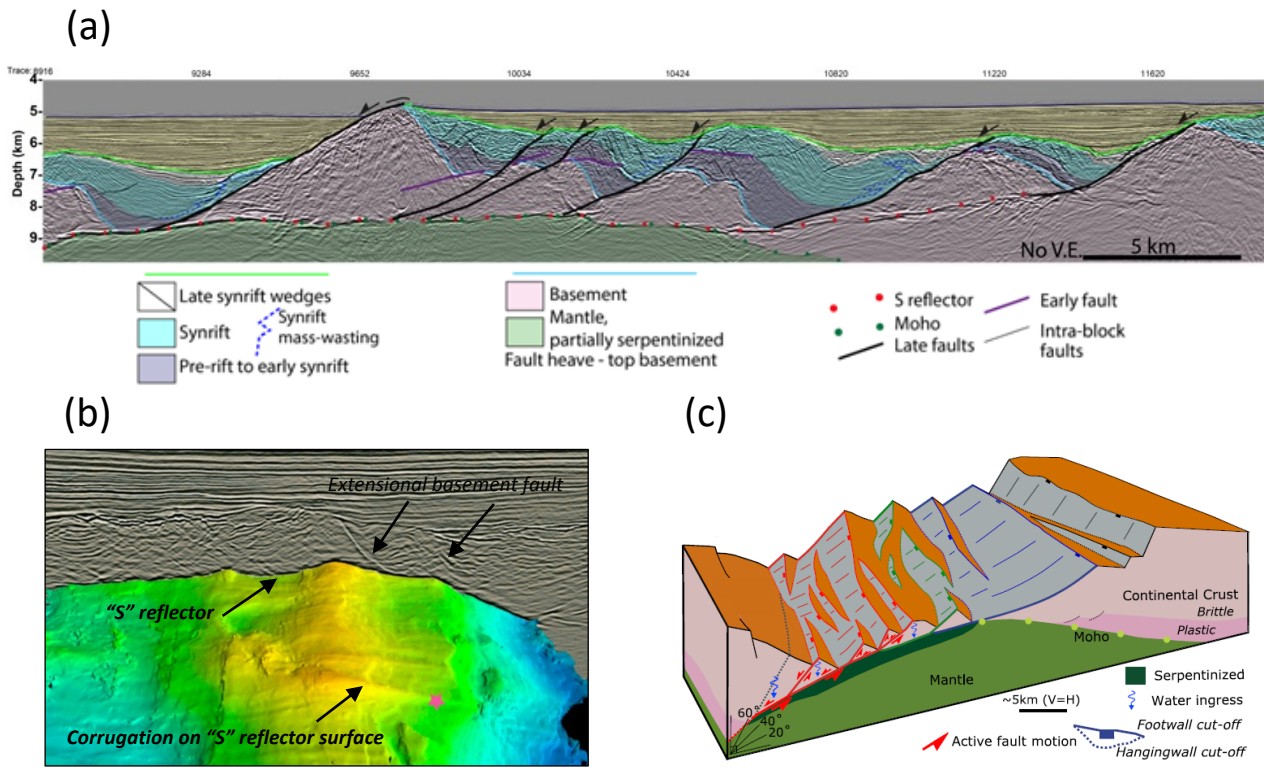

**Figure 1: a)** Depth seismic reflection section across the hyper-extended domain of the SW Galica Bank continental margin showing the relationship between basement extensional faults, the "S" horizontal detachment and syn- and post-tectonic sediment fill (modified from Figure 5b of Lymer et al, 2019). **b)** 3D view extracted from a 3D seismic reflection cube in hyper-extended domain of the Porcupine Basin, showing a seismic line and the interpreted "S" reflector surface in two-way travel time (adapted from Figure 2b of Lymer et al, 2022). It illustrates the horizontal detachment corrugations and their relationship with the extensional basement faults above. **c)** Summary schematic model of extensional faulting within the hyper-extended domain of the Iberia magma-poor rifted margin based on 3D seismic reflection interpretation (Lymer et al. 2019).

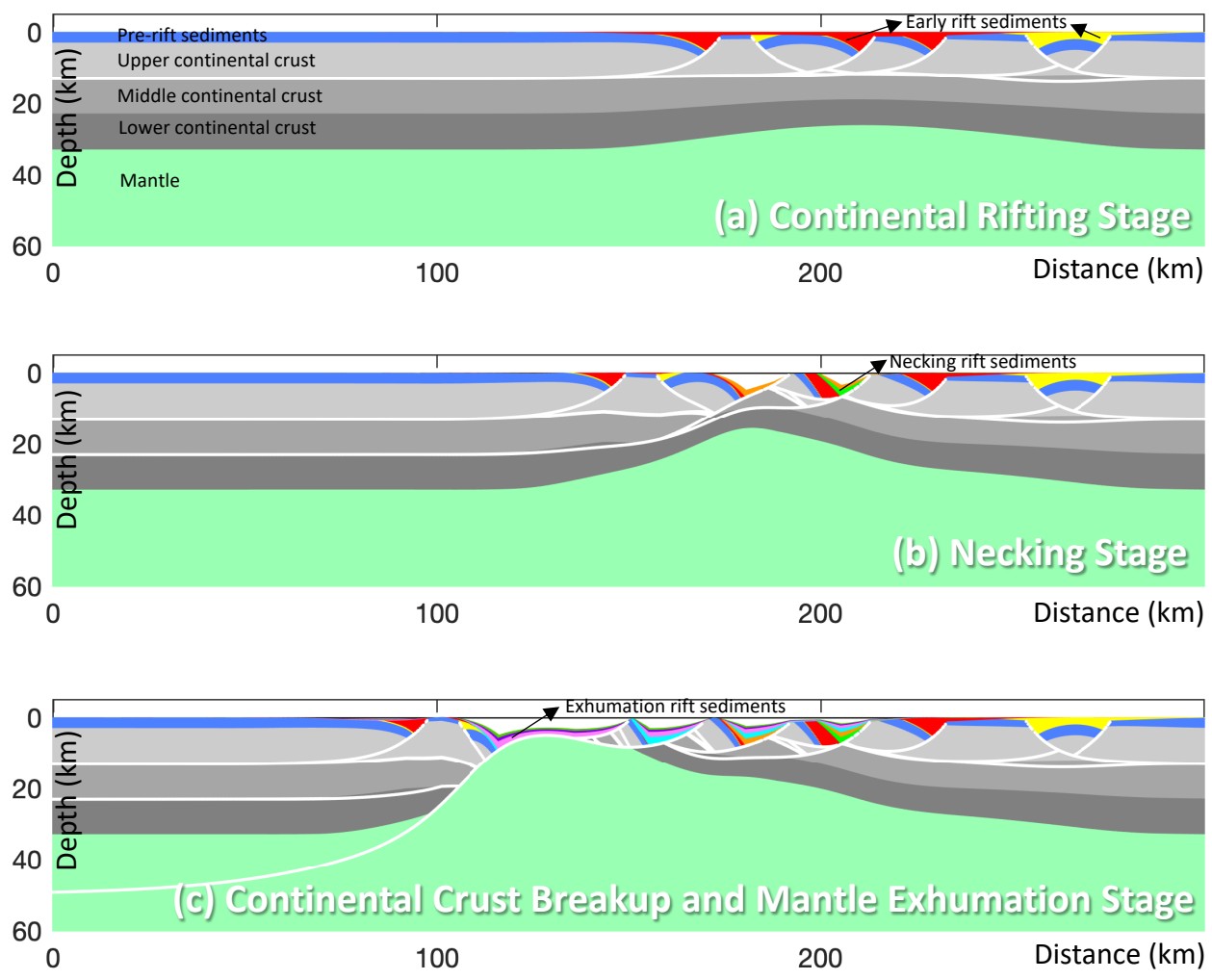

**Figure 2:** Example application of the kinematic lithosphere deformation model (RIFTER) applied to magma-poor rifted margin development: **a) c**ontinental rifting stage, **b) n**ecking stage, **c)** crustal breakup and mantle exhumation stage. The model computes the flexural isostatic response to changes in lithosphere loading including the rolling hinge flexural rotation process during extensional faulting.

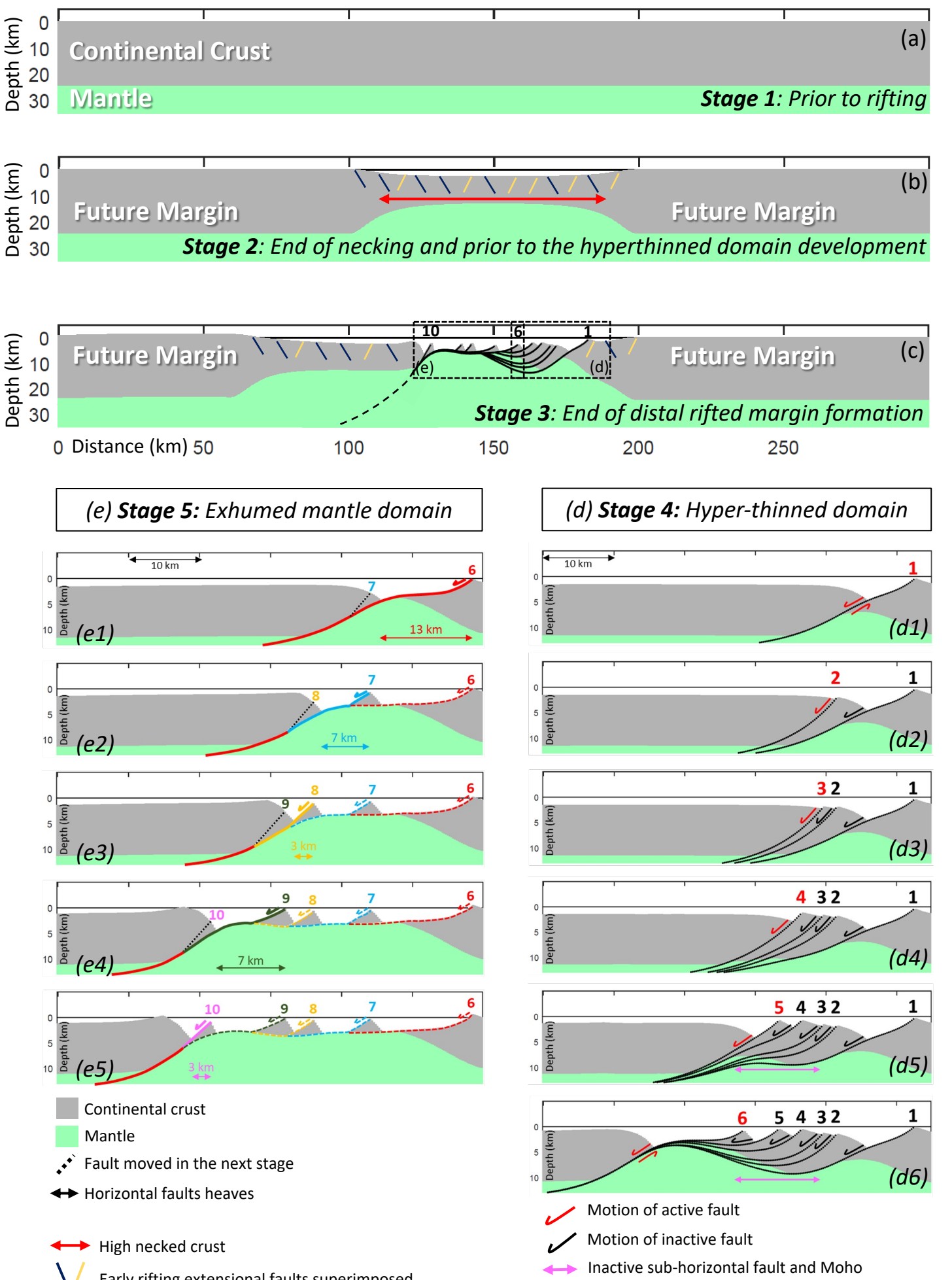

**Figure 3:** A generalized evolutionary RIFTER model showing the development of a magma-poor rifted margin. **a)** Lithosphere architecture prior to rifting. **b)** Lithosphere architecture at the end of the necking stage, prior to the formation of hyper-extended domain. **c)** Formation of hyper-thinned domain by in-sequence oceanward extensional faulting leading to mantle exhumation. **d)** Detail of the hyper-thinned domain formation (d1-d6). **e)** Detail of the exhumed mantle domain formation (e1-e5).

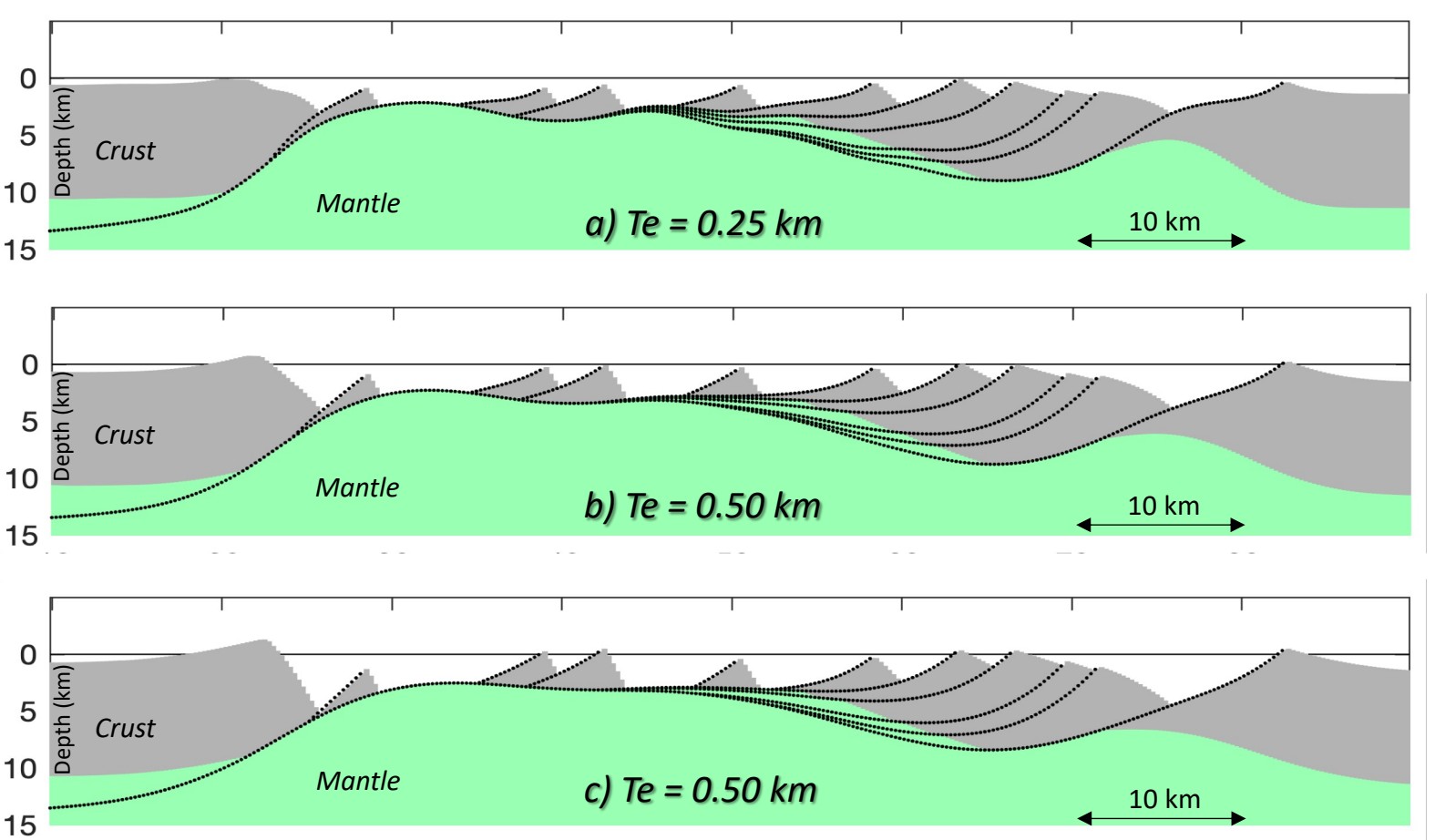

**Figure 4:** Model sensitivity to the effective elastic thickness, Te, used to determine the flexural response to extensional faulting. Fault location, extension, initial dip and activation sequence are the same as in Figure 3c.

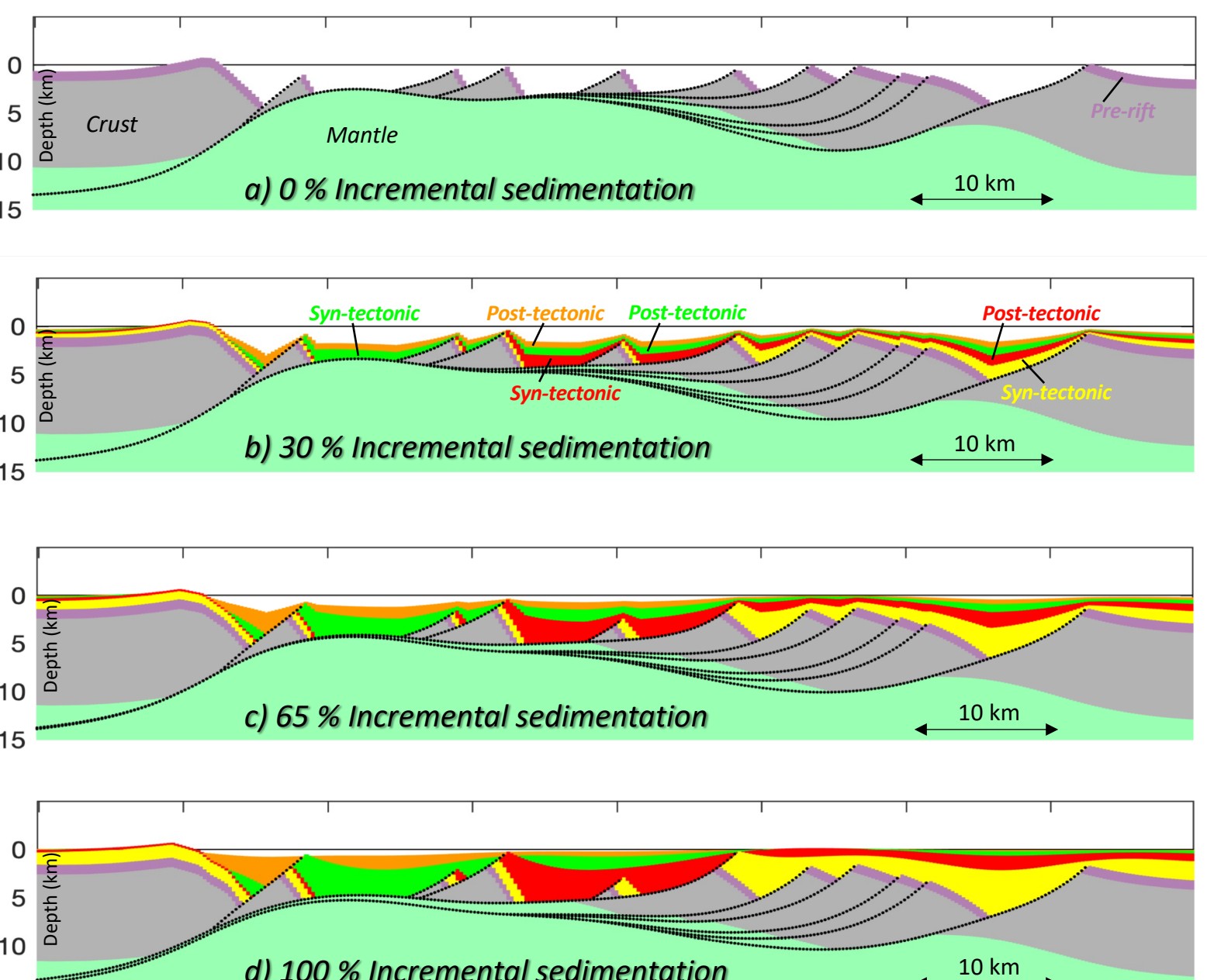

**Figure 5:** Sensitivity to sediment supply of a generalized evolutionary RIFTER model (same tectonics as in Figure 3e) with incremental sediment deposition during oceanward in-sequence extensional faulting. Sediment supply is parameterised as the % of available accommodation space filled by sediment prior to the isostatic response to sediment loading. Diachronous oceanward in-sequence extensional faulting results in sediment packages of the same age being syn-tectonic or absent distally (to left) but post-tectonic proximally (to right). Sediment isostatic loading is included but sediment compaction is not.

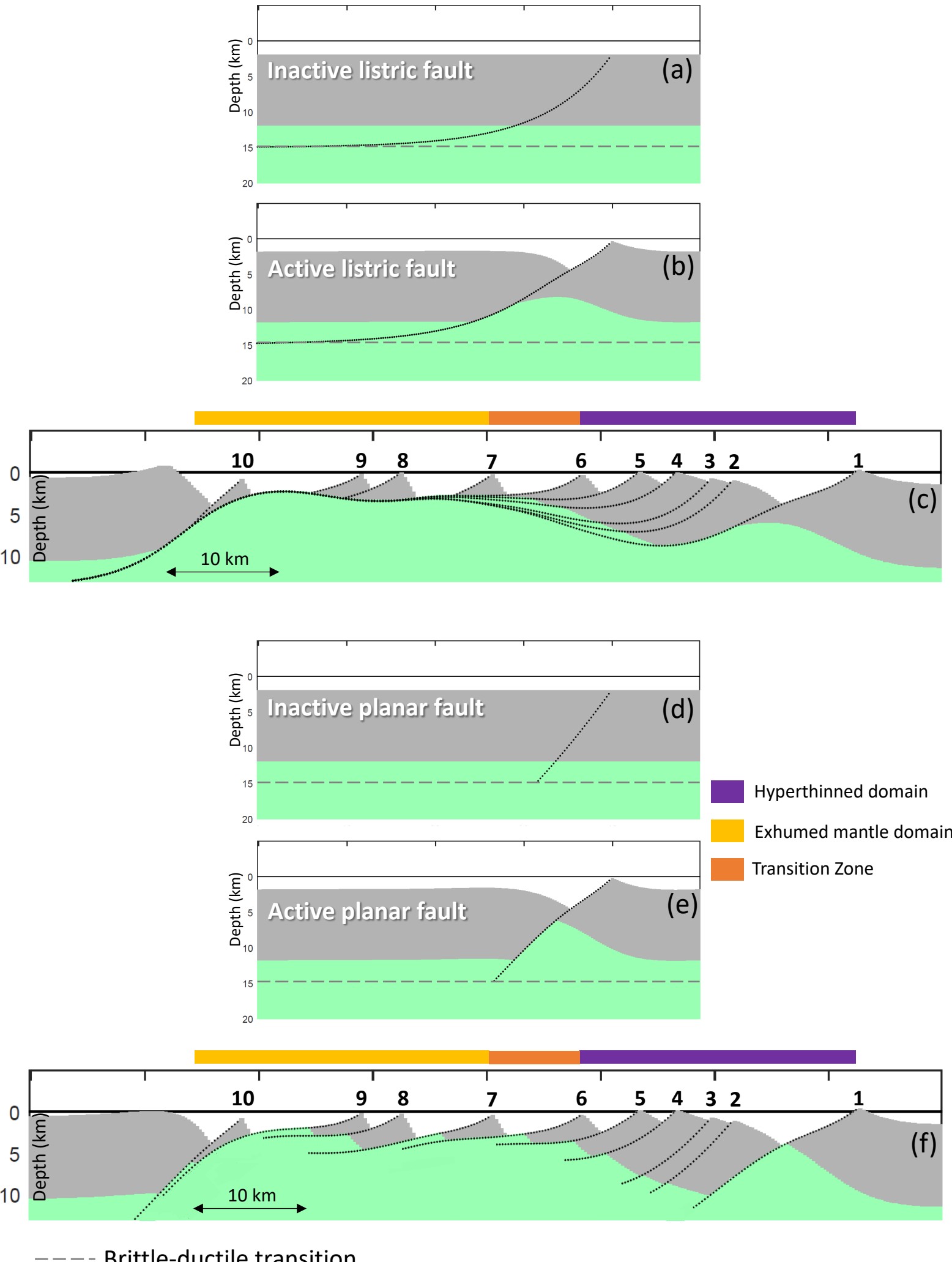

**Figure 6:** Comparison of hyper-extended domain structure and transition to exhumed mantle predicted using listric and planar faults in the RIFTER model. a-c) Using listric faults (same as shown in Figure 3c) and d-f) using planar faults.

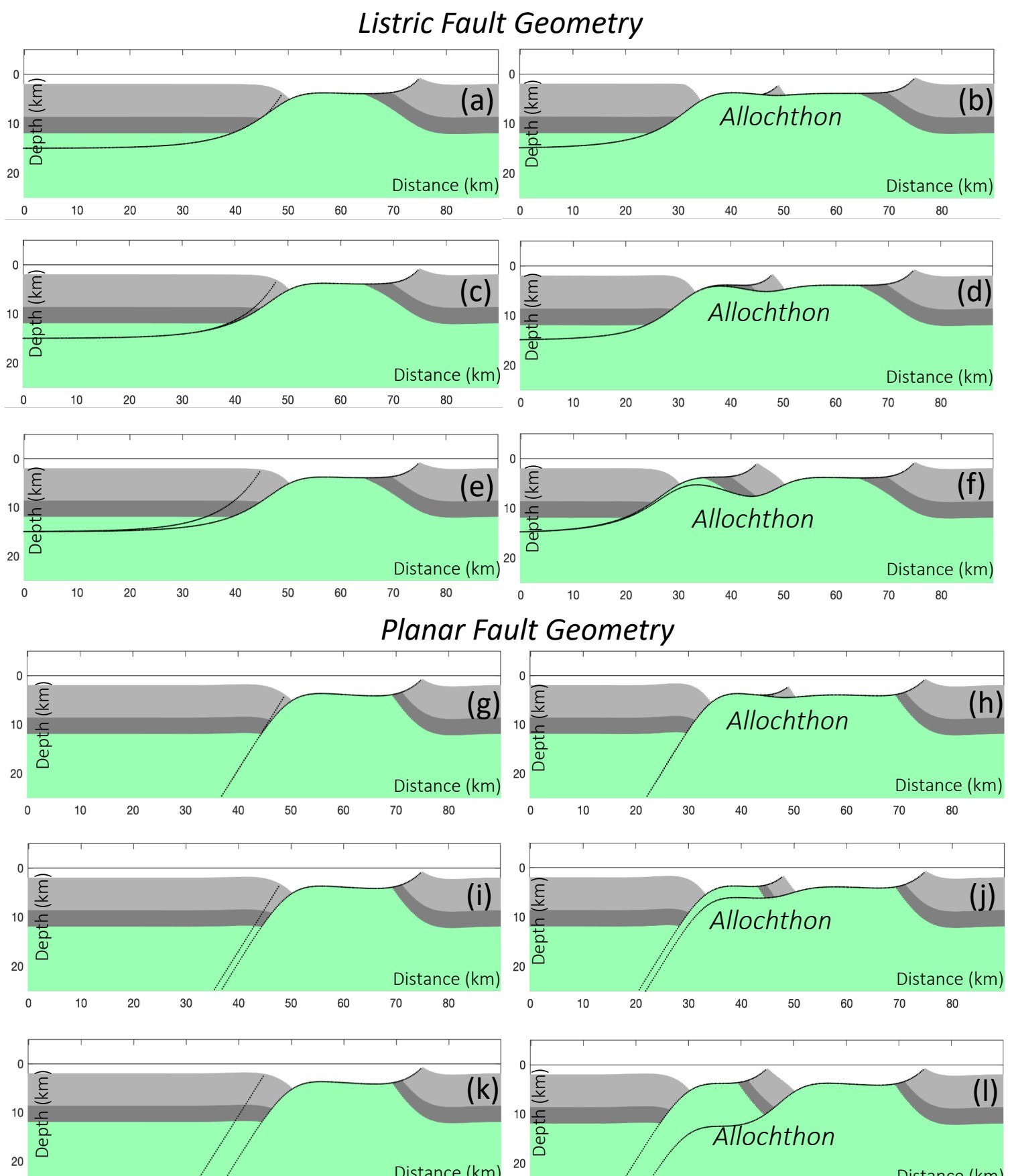

**Figure 7:** Comparison of allochthon block formation using listric (a-f) and planar (g-l) fault geometry for different offsets of new short-cut fault with respect to footwall emergence of primary fault. Initial fault dip 60°, detachment depth = 15 km for listric fault, Te = 0.5 km. a, b, g & h) 1 km offset of new short-cut fault with respect to footwall emergence of primary fault before and after 15 km of extension and predicted extensional allochthon block for listric and planar fault geometry. c, d, i & j) corresponding model prediction with 2 km offset. e, f, k & l) corresponding model prediction with 5 km offset.

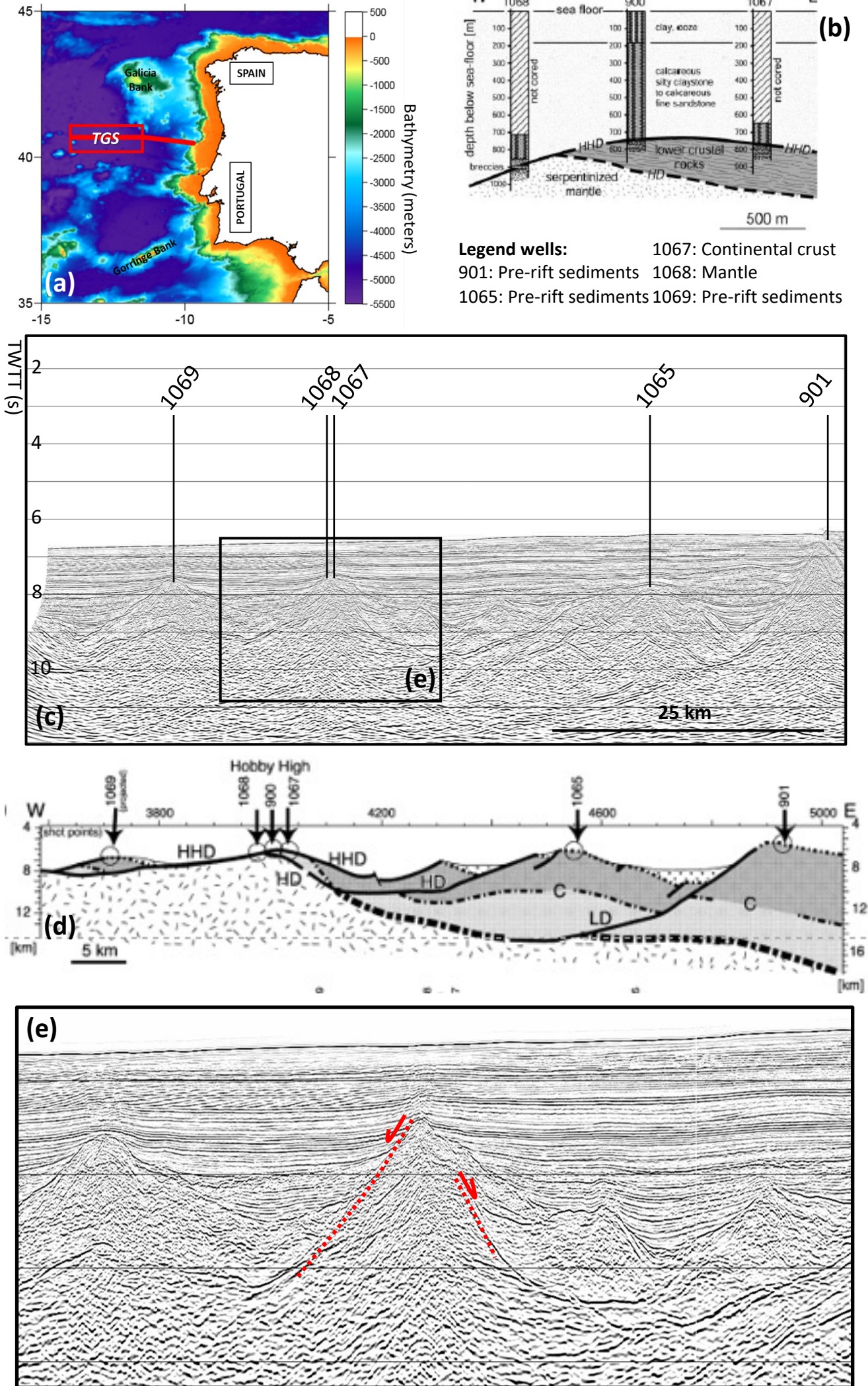

**Figure 8:** a) Bathymetric map of the western Iberian margin showing in red the location of TGS seismic reflection profile. b) ODP well observations from the western Iberia margin (Manatschal et al., 2001 and 2004). c) Part of the TGS time domain seismic reflection section (Sutra and Manatschal et al., 2012) showing ODP well locations (black lines). d) Interpretation of the above by Manatschal et al., (2001 and 2004). e) Interpretation of out-of-sequence faulting for inset of seismic section shown in c).

## (a) In-sequence scenario: master fault (number 6)

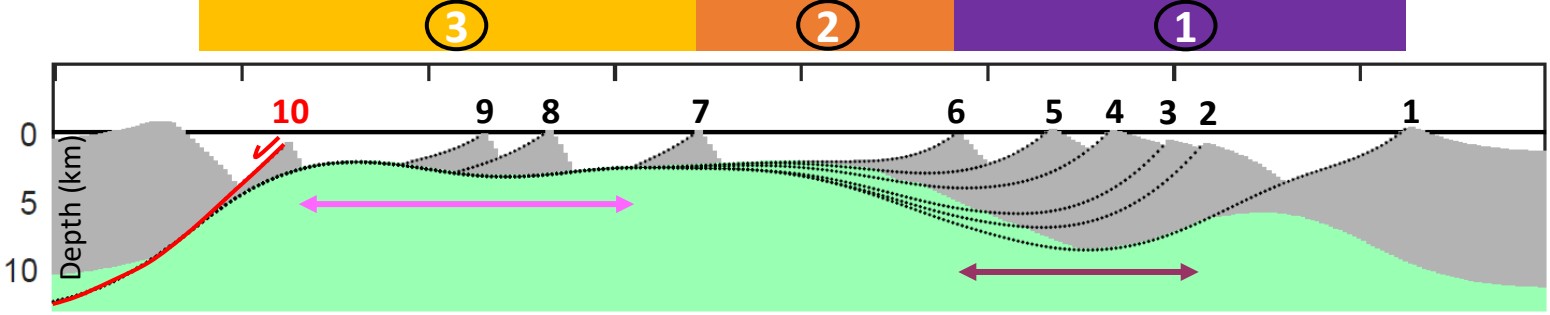

## (b) Out-of-sequence scenario: ocean-dipping fault (O)

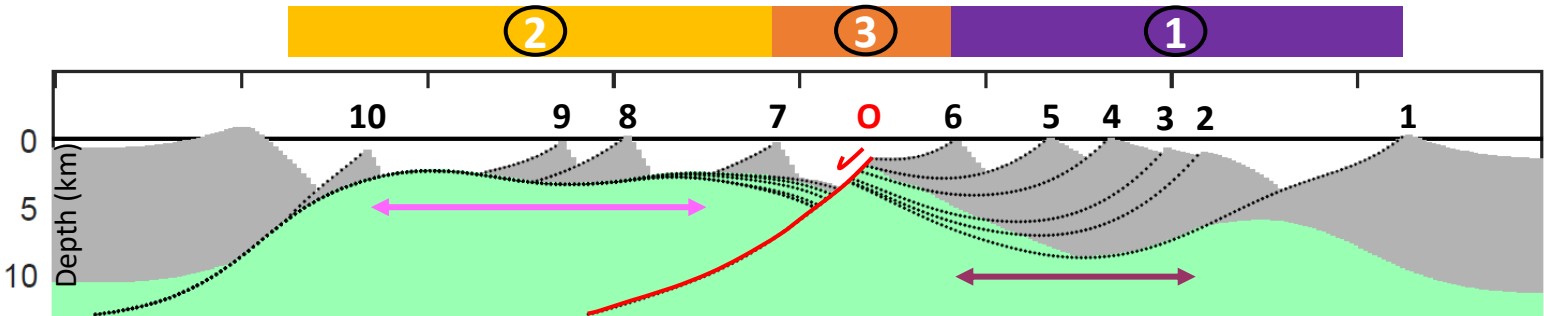

## (c) Out-of-sequence scenario: ocean- and continent-dipping fault (O and C)

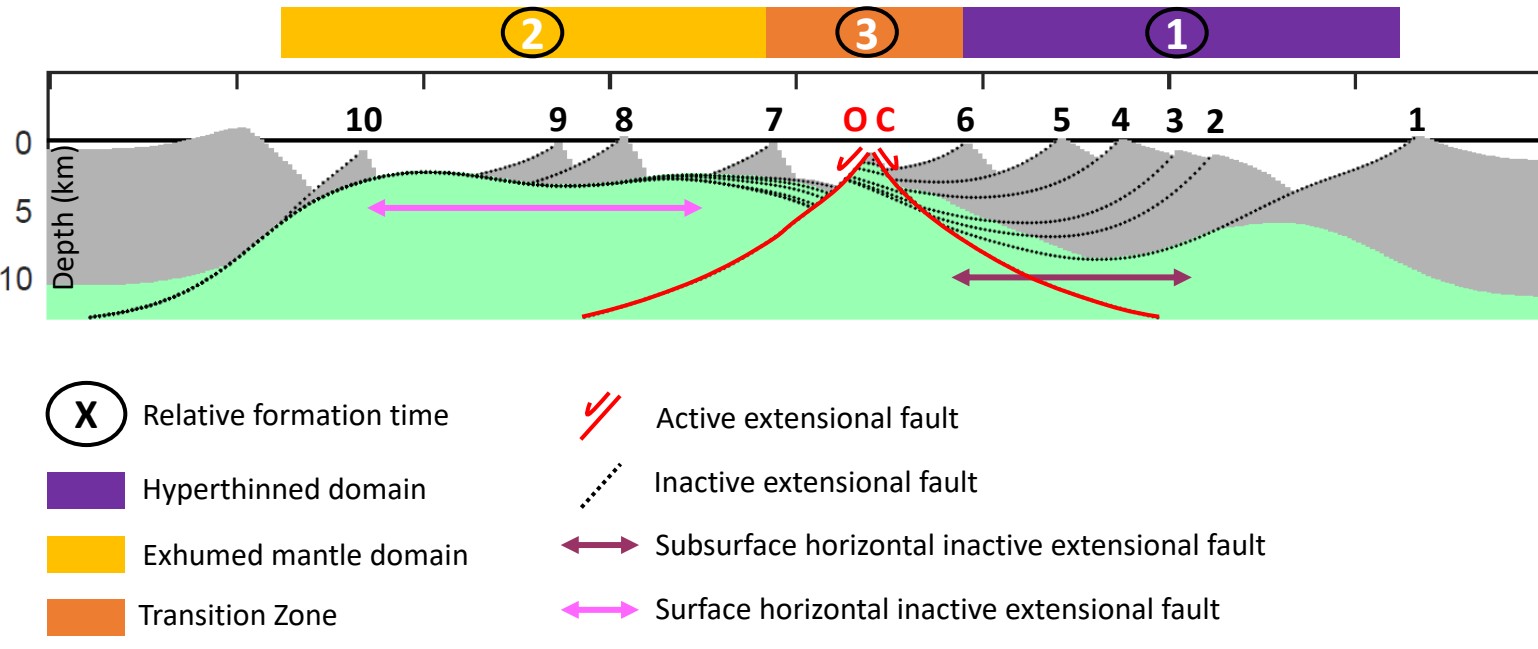

**Figure 9:** Comparison of predicted transition from hyper-extended crust onto exhumed mantle for in-sequence and out of sequence faulting. Crust and mantle are grey and green respectively. a) In-sequence faulting produces a smooth bathymetric transition from hyper-extended crust to exhumed mantle. b & c) Out of sequence faulting produces a transition from hyper-extended crust to exhumed mantle with bathymetric relief.

| Faults numbers | 1 | 2 | 3 | 4 | 5 | 6 |
|---|---|---|---|---|---|---|
| Horizontal faults heaves (km) | 7 | 0,5 | 0,5 | 1,5 | 4 | 13 |
| Initial fault dip (listric fault) | Surface = 60° | | | | | |
| | At 15 km = 0° | | | | | |
| Fault movement | Red number = fault active | | | | | |
| | Black number = fault inactive | | | | | |

**Table 1:** Table for fault parameters used for Figure 3d. Fault number indicates the chronological movement (Fault 1 is the oldest).

| Faults numbers | 6 (master fault) | 7 | 8 | 9 | 10 |
|---|---|---|---|---|---|
| Horizontal faults heaves (km) | 13 | 7 | 3 | 7 | 3 |
| Initial fault dip (listric fault) | Surface = 60° | | | | |
| | At 15 km = 0° | At 30 km = 0° | | | |
| Fault movement | Colour solid line = fault active | | | | |
| | Colour dash line = fault inactive | | | | |

**Table 2:** Table for fault parameters used in Figure 3e. Fault number indicates the chronological movement (Fault 6 is the oldest).

| Domains formed | 1. Hyperthinned domain | 2. Transition zone | 3. Exhumed mantle domain |
|---|---|---|---|
| **Horizontal faults heaves (km)** | Faults numbers | 6 (master fault) | |
| | Horizontal faults heaves (km) | 13 | |
| | Initial fault dip (lisitric fault) | Surface = 60° | |
| | | Depth at 15 km = 0° | |

**Table 3:** Table for fault parameters used in Figure 7a.

| Domains formed | 1. Hyperthinned domain | 2. Exhumed mantle domain | 3. Transition zone |
|---|---|---|---|
| **Horizontal faults heaves (km)** | Faults numbers | 6 (master fault) | Fault O (ocean) |
| | Horizontal faults heaves (km) | 7 | 2 |
| | Initial fault dip (lisitric fault) | Surface = 60° | Surface = 60° |
| | | Depth at 15 km = 0° | Depth at 15 km = 0° |

**Table 4:** Table for fault parameters used in Figure 7b.

| Domains formed | 1. Hyperthinned domain | 2. Exhumed mantle domain | 3. Transition zone |
|---|---|---|---|
| **Horizontal faults heaves (km)** | Faults numbers | 6 (master fault) | Fault O (ocean) | Fault C (continent) |
| | Horizontal faults heaves (km) | 7 | 2 | 1 |
| | Initial fault dip (lisitric fault) | Surface = 60° | Surface = 60° | Surface = 60° |
| | | Depth at 15 km = 0° | Depth at 15 km = 0° | Depth at 15 km = 0° |

**Table 5:** Table for fault parameters used in Figure 7c.