# Peer review of "Extensional fault geometry and evolution within rifted margin hyper-extended continental crust leading to mantle exhumation and allochthon formation"

_EGUsphere, 2023_

## Author Response (AR1)

**Reviews comments**

**Response to reviews comments**

Hi, this comment is a bit of a cop-out I'm afraid. Having read the paper, and realised that is 95% computer modelling-based, I feel I am probably not the right person to see this one through the whole review process. I've worked with computer tectonic modellers, but never done it myself. Therefore you need someone who can properly examine the input variables, boundary conditions, iterations, etc. May I suggest John Naliboff if you haven't already asked him? I worked with him on a recent paper and he is excellent at this stuff. He's at New Mexico Tech, john.naliboff@nmt.edu.

However, as this is an open review anyway, a few general comments:

1) Most importantly, you need to describe why this is an important (or at least useful) contribution to the discussion on the structure of highly extended margins. How does it differ from the models already published, or does it just confirm and add weight to them via lithospheric modelling?

2) It would be useful to define in simple terms, up front, what "in sequence" and "out of sequence" faulting mean. Not everyone reading this will have been as deeply immersed in the debate as you are, so some definition of terms would help the general reader.

3) In my opinion, there isn't enough actual seismic in this paper. It's almost all computer output, with nothing to tie it to. Apart from Fig. 6 (which actually describes a special case) it's just assumed that the reader has seen the seismic elsewhere. For completeness - and real-world grounding, please consider at least one more general seismic line over the margin.

All general comments, I know. You need an expert modeller for the rest.

Best wishes and good luck with the process,

Tony Doré

Energy & Geoscience Institute (U. Utah affiliated, but based in London)

October 2023

We thank Tony Dore for his 3 comments. We understand and agree with them – they are constructive and appropriate.

The aim of our paper is to investigate whether the rolling hinge fault model of Buck, used very successfully to explain fault processes at magma-poor slow-spreading

ocean ridges, can reproduce the evolution of extensional faulting interpreted from the 3D seismic imaging of the hyper-extended domain of the Iberian magma-poor rifted margin. The 3D seismic analysis by Lymer et al. (2019), following earlier 2D seismic work (e.g. by Reston, Ranero, Pérez-Gussinyé), has, we believe, answered long standing questions concerning the relationship between high-angle extensional faulting and the "S" sub-horizontal reflector.

As well as making the aims of our paper clearer, we need to summarise better the extensional faulting process proposed by Lymer at al. from their 3D seismic analysis, perhaps by showing an example of their seismic sections in our figure 1. We also need to explain what is meant by in-sequence extensional faulting as used in the papers we cite as well as our own paper.

The paper by Gomez-Romeu and Kusznir presents modelled fault geometries, whose sequential restoration includes the effects of flexural isostacy. The main aim is to demonstrate how different starting fault geometries (listric vs planar faults, fault angle) evolve with time, and respond to isostatic effects, and consequently, what criteria might be identified to discriminate the different starting geometries on seismic reflection data. The paper is well written and referenced, and the illustrations are good. The examples presented usefully illustrate the basic range of geometries that can be expected for large displacement faults on hyper extended margins, and how the sequence of deformation is important. I only have minor comments – below.

Lines 111-113 Perhaps could be more specific about the starting configuration. Fig. 3 – was a) and b) modelled by pure shear and the conjugate faults are just schematically illustrated?

While Fig. 1 is a useful summary, it would be good to have at least one regional seismic line that illustrates the full regional picture of what you are modelling. Because Iberia is referred to numerous times in the text, it got me wondering if you were actually trying to   model a particular part of the margin. I think it would be helpful to more clearly state somewhere around lines 52-59 that you are not modelling a specific margin or seismic line, but you have created generic models to address the types of feature found along the Iberia margin (and perhaps cite other examples too).

Model formulation – although the model can include sedimentation, it does not appear that sedimentation has been built into the modeling, and I did not see in section 2 any mention about why this has been omitted. I presume it is because the syn-kinematic sediments are likely to be thin, and not much denser than the water column. But I think this simplification of the model should be addressed. It also provides the opportunity to address your assumptions about how rapidly you think the extension would proceed (slow extension would acquire more syn-kinematic sediment).

I accept that these margins can largely evolve from normal faults that dip at typical normal fault angles (i.e. 45-60 degrees), so have no issue with the scenarios presented here. There is an implication that variations in starting fault angle do not need to be considered. However, I think it would be interesting to look at normal faults with lower starting angles (20-30 degrees) as well, just to see what differences may arise, and are features produced that are incompatible with the seismic data. Reactivation of low-angled zones of weakness, such as thrusts, may sometimes produce initially low-angle normal faults.

One simple way to estimate the initial angle of a fault is to assume bedding was initially horizontal, and use the bedding-fault cutoff angle to determine the initial fault angle. It would be useful if you could comment on whether there are significant changes in cutoff angle as your model increases displacement, and what magnitude of variations occur.

 Summary, perhaps being more specific about the nature of the S reflection and how your modelling of planar and listric faults differ would be useful.

I enjoyed reading the manuscript.

Chris Morley

We thank Chris Morley for his comments and suggestions.

We agree that adding a regional seismic reflection section to figure 1 would improve the accessibility of the paper to readers and also illustrate our review and summary of what has been learnt by seismic reflection analysis in the hyper-extended domain of the Iberia continental margin. We also agree that we should make clear that we are not modelling a particular seismic line but rather that we are modelling the generic processes operating during the formation of an hyper-extended rifted magma-poor margin.

The model that we use can incorporate sediment deposition during the incremental tectonic development of a magma-poor rifted margin. We did not include sedimentation in order to focus on the tectonics and fault evolution – however we should perhaps reconsider this. We can include an additional figure showing this which would illustrate the diachronous nature of fault extension and the distinction between syn- and post-tectonic sedimentation.

We note your comment that we only consider normal faults with steep initial angles (~60°). We believe that this starting angle is appropriate to the Iberian margin and many others. We agree however, that where extensional faulting reactivates thrust faults, that the starting angle is much lower (30° or less). A good example of this is on the SW margin of the South China Sea where thrusts are reactivated as extensional fault. Preliminary modelling of reactivated low angle thrusts shows that the flexural

response to extensional faulting successfully reproduces observations and is sensitive to fault angle. This is work in progress and beyond the scope of this paper which focuses on fault geometry evolution during the hyper-extension of magma-poor rifted margins.

We believe that the question of whether the bedding-fault angle can be used to determine the initial fault angle has also already been addressed in papers summarising seismic reflection observations at hyper-extended rifted margins. We will add text and references to summarise this.

Regarding listric versus planar fault geometry, we believe that the shallower parts of extensional fault during hyper-extension are planar but become listric as they sole into the S reflector. This is an important point and we will make it clearer in the revised text.

---

## Author Response (AR2)

**Response to Comments and Recommendations of the Editor (Mohamed Gouiza)**

We thank Mohamed for his comments (in italics below). We have updated the text in response to his suggestions. A summary of our responses is given below.

*Chris raised an interesting question regarding the effect of sedimentation on the modelling results. This was addressed in the revised manuscript (Lines 203-210 and Figure 4), which suggests that sedimentation has no effect on the structural evolution of the model. One would expect the opposite, but I presume that since the syn-kinematic sequences in these domains (i.e., hyperextended and exhumed mantle) are often thin, the effect of sedimentation would be negligible anyway.*

Our updated text in lines 238 to 246 regarding sediment loading of the model

The model results of increasing sediment supply are shown in Figures 5b-c and compared with the model result with no sediment deposition shown in Figure 5a. Figure 5b shows a relatively small amount of sediment incrementally added to the model and is consistent with a relatively sediment starved scenario corresponding to the SW Galicia margin as imaged by the 3D seismic of Lymer et al (2019). The isostatic response to the small amount of sediment loading shown in Figure 5b is also small and the flexural isostatic fault rotation is therefore not significantly different from the model result with no sediments shown in Figure 5a. The increased isostatic response to increasing sediment supply (Figures 5c&d) results in a slight decrease in fault rotation resulting in slightly steeper faults for the same fault extension. Sediment supply and its isostatic loading are therefore expected to exert a control on when faults lock and new oceanward in-sequence faults develop.

*However, I am wondering if this has to do with the numerical modelling approach as well? This is why, I think it is essential to address the limitation of the numerical model RIFTER. You do refer the reader to published literature where detailed description of the model formulation is provided, but I think you should address the impact of the assumptions and simplifications of the model formulation on this particular case study. For instance, the lack of a dynamic implementation of temperature and the assumed initial Te of 0.5km. This could be addressed either in the discussion or in a separate section just before the discussion.*

Our updated text in lines 101 to 125 regarding the kinematic model

We use a numerical model (RIFTER) to replicate faulting and fault block geometry within the hyper-extended domain, and to investigate fault rotation, fault geometry interaction, the formation of crustal allochthon blocks and the transition between hyper-extended and exhumed mantle domains. RIFTER is a kinematic forward lithosphere deformation model that allows the production of flexural isostatically compensated as well as balanced cross-sections. Within RIFTER, lithosphere is deformed by faulting in the upper crust with underlying distributed pure-shear deformation in the lower crust and mantle. RIFTER can be used to model and predict the structural development in extensional tectonic settings as shown in Figure 2. The model is kinematically controlled with fault geometry, fault displacement and pure-shear distribution given as model inputs as a function of time.

The kinematic formulation of RIFTER represents an advantage over dynamic modelling because the input data given to RIFTER can be constrained by observed geology. Specifically fault position, extension magnitude and sequence order with respect to other faults can be taken directly from the interpretation of seismic reflection images and used to drive the kinematic model. This is in contrast to dynamic models where fault location, extension magnitude and sequence order are predicted by the model and may bare little relationship to an observed structural and stratigraphic cross-section. In

a kinematic model, while the lithosphere deformation is specified as an input, the thermal and isostatic consequences may be dynamically determined to predict thermal uplift and subsidence (e.g. Gómez-Romeu et al. 2019). Because model outputs are geological cross-sections which are flexural isostatically compensated as well as structurally balanced, RIFTER provides for the isostatic testing of palinspastic cross-sections and can also be used to explore different kinematic scenarios. A more detailed description of the model formulation (originally called OROGENY) is given by Toth et al., (1996), Ford et al., (1999) and Jácome et al., (2003). These studies show the model formulation applied to compressional tectonics however similar physical principles apply for an extensional tectonics scenario. Gómez-Romeu et al., (2019) show how RIFTER can be used to reproduce both extensional and compressional tectonics using the Western Pyrenees as a case-study.

Our updated text in lines 141 to 147 regarding choice and sensitivity to Te

We use a Te value of 0.5 km in our modelling of extensional faulting during the formation of the hyperextended domain and mantle exhumation (Figure 3).  This value is consistent with those determined at slow-spreading ocean ridges ranging between 0.5 and 1 km (e.g. Buck, 1988; Smith et al., 2008; Schouten et al., 2010) where a similar lithosphere flexural strength to that of the distal rifted margins is expected. The sensitivity of model predictions to Te is shown in Figure 4; increasing Te increases the bathymetric relief.resulting from extensional faulting but otherwise the structural architecture remains similar.

*I also believe that there is an important point that was raised by Tony that was not considered in the revised manuscript: How does your contribution differ from the models already published? and I would add: How does you modelling results compare to what is already published? There is at least one published work that I can think of that also addressed the process of hyperextension and mantle exhumation in rifted margins, but using geodynamic modelling, by Peron-Pinvidic & Naliboff (2020, https://doi.org/10.1130/G47174.1).*

Our updated text in lines 72 to 93 regarding relation of our work to previous studies

Dynamic thermo-rheological finite element models of continental lithosphere stretching and thinning (e.g. Lavier & Manatschal, 2006; Brune et al. 2014; Naliboff et al. 2017) leading to continental breakup and rifted margin formation have been successful in simulating the progression from necking to hyper-extension to mantle exhumation. at magma-poor rifted margins. However these dynamic models do not replicate the extensional fault and detachment structures observed on 2D and 3D seismic reflection data. The dynamic model of Peron-Pinvidic & Naliboff (2020), specifically investigating extensional detachment development, predicts extensional fault structures that penetrate to depths much greater than the seismically observed S-type reflector; additionally their predicted fault geometries remain steep failing to match the lower fault angles imaged on seismic reflection data. The kinematic model presented by Ranero & Perez-Gussinye (2010) using extensional fault block rotation much better replicates extensional fault and detachment structures imaged by 2D seismic within the hyper-extended magma-poor margin domain. Their work however preceded the 3D seismic observations by Lymer et al (2019) of the S-type detachment and its corrugations.

Lymer et al. (2019) propose that their observations strongly support the development of the S seismic reflector by a rolling-hinge process (Buck 1988) in which a sub-horizontal detachment is created by the incremental addition of the soles of basement extensional faults. The kinematic rolling-hinge model of Buck (1988) has been successfully used at slow spreading ocean ridges to replicate and analyse extensional faulting leading to footwall exhumation, detachment faulting and core complex formation (Smith et al. 2008; Schouten et al, 2010). In this paper, we use a recursive adaptation of the rolling

hinge model of Buck (1988) to examine how both active and inactive fault geometries are modified by flexural isostatic rotation during sequential faulting to form the sub-horizonal structure imaged on seismic reflection data.

*One last minor suggestion regarding the caption of Figure 1c, which could be improved as follow: 3D view extracted from a 3D seismic reflection cube in hyper-extended domain of the Porcupine Basin, showing a seismic line and the interpreted "S" reflector surface in two-way travel time (adapted from Figure 2b of Lymer et al, 2022). It illustrates the horizontal detachment corrugations and their relationship with the extensional basement faults above.*

We have updated the caption for Figure 1b with the text above.

Julia Gomez Romeu and Nick Kusznir